# FreeEnhance: Tuning-Free Image Enhancement via Content-Consistent Noising-and-Denoising Process

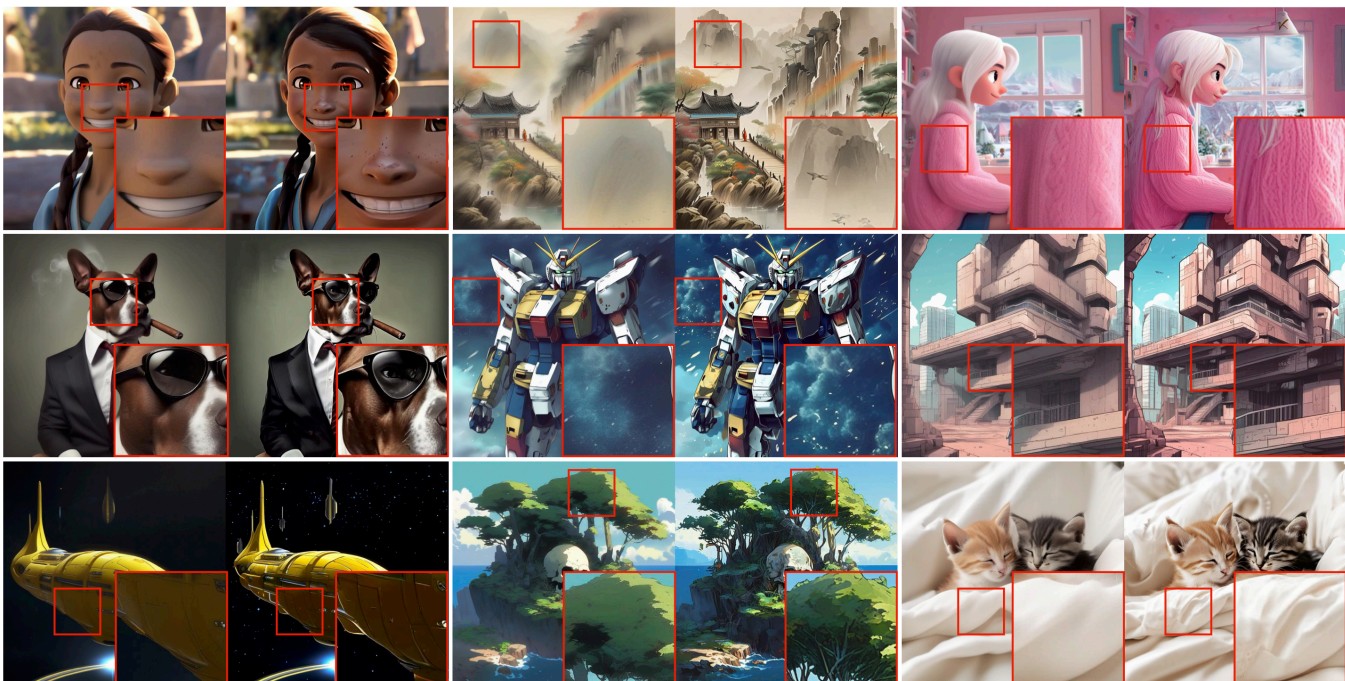

Figure 1: The landscape examples of FreeEnhance versus SDXL. In each pair of images, the left one is generated by SDXL at a resolution of $1,024 \times 1,024$, while the right one is produced by FreeEnhance using the SDXL-synthesized image as the input. FreeEnhance preserves the resolution of the input images while introducing additional details in a content-consistent manner.

## ABSTRACT

The emergence of text-to-image generation models has led to the recognition that image enhancement, performed as post-processing, would significantly improve the visual quality of the generated images. Exploring diffusion models to enhance the generated images nevertheless is not trivial and necessitates to delicately enrich plentiful details while preserving the visual appearance of key content in the original image. In this paper, we propose a novel framework, namely FreeEnhance, for content-consistent image enhancement using the off-the-shelf image diffusion models. Technically, FreeEnhance is a two-stage process that firstly adds random noise to the input image and then capitalizes on a pre-trained image diffusion model (i.e., Latent Diffusion Models) to denoise and enhance the image details. In the noising stage, FreeEnhance is devised to add lighter noise to the region with higher frequency to preserve the high-frequent patterns (e.g., edge, corner) in the original image. In the denoising stage, we present three target properties as constraints to regularize the predicted noise, enhancing images with high acutance and high visual quality. Extensive experiments conducted on the HPDv2 dataset demonstrate that our FreeEnhance outperforms the state-of-the-art image enhancement models in terms of quantitative metrics and human preference. More remarkably, FreeEnhance also shows higher human preference compared to the commercial image enhancement solution of Magnific AI.

## KEYWORDS

Image Generation, Image Enhancement, Diffusion Model

## 1 INTRODUCTION

The recent development of diffusion models has sparked a remarkable increase in research area for the purpose of generating multimedia content. Among these endeavors, text-to-image generation stands out as one of the most representative tasks [14, 57]. Diffusion Probabilistic Models (DPM) [20, 36, 46] approaches image generation as a multi-step denoising process, employing a powerful

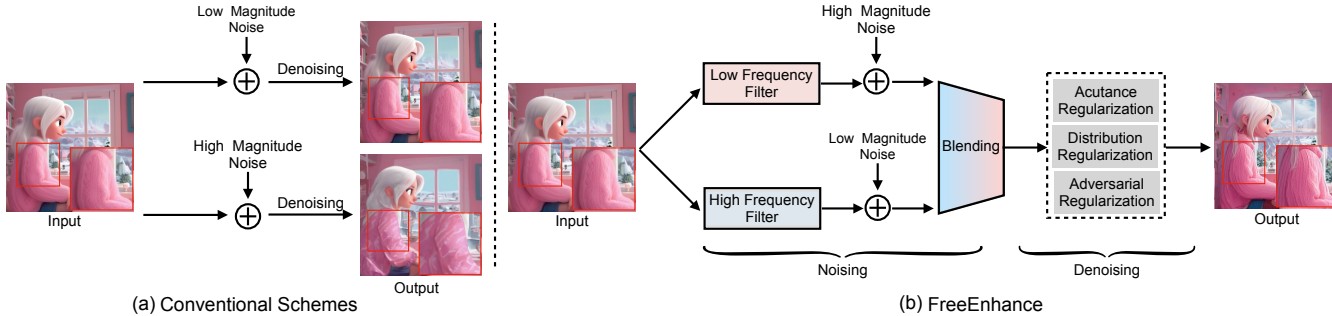

**Figure 2: The conventional image enhancement via (a) noise-and-denoising pipeline suffers from tradeoffs between creativity and content-consistency. We introduce (b) FreeEnhance, a tuning-free framework that selectively adds lighter noise in high-frequency regions to preserve content structures, while heavier noise is added in low-frequency regions to enrich details in smooth areas. Moreover, three regularizers are employed to further improve visual quality during denoising.**

denoiser network to progressively transform a Gaussion noise map into an output image. Building upon this method, Latent Diffusion Models (LDM) [38, 41] propose to execute denoising process in the latent feature space that is established by a pre-trained autoencoder, leading to high computation efficiency and image quality. More recently, to improve the controllability of text-to-image generation, ControlNet [57] and T2I-Adapter [34] incorporate various spatial conditions into the denoiser network. Despite showing impressive progress in content controlling, synthesizing high-quality image remains challenging, due to the lack of visual details in the generated images, as shown in the Figure 1.

To enrich details in the generated images, one general solution is the "noising-and-denoising" process, which first properly adds noise to the original image, and then exploits a diffusion model to denoise the noisy image. This idea is originally proposed in SDEdit [31] for image editing, and then explored in SDXL [38] to enhance the generated image, as illustrated in Figure 2(a). Nevertheless, the effectiveness of such process highly relies on the strength of the attached noise. Specifically, when the noise magnitude is low, the input image cannot be effectively enhanced, whereas when it is high, the key content (e.g., human or objects) undergoes significant changes that deviate from the original input image. To alleviate this limitation, we propose to remould this process by selectively adding lighter noise in high-frequency regions to preserve edge and corner details, while heavier noise is added in low-frequency regions to carry more details in the smooth area, as shown in Figure 2(b). Moreover, we devise three types of regularizations to correct the denoising process and produce images with superior acutance and visual quality.

To materialize our idea, we propose a new FreeEnhance framework, that remould the standard noising-and-denoising process to improve the visual quality of the input image and meanwhile keep the key content consistent. Firstly, we divide the input image into high-frequency and low-frequency regions by utilizing a high-pass filter. For the high-frequency region, we employ DDIM inversion [33] to attach light noise, which is easier to be eliminated by using a diffusion model than random noise. For the low-frequency region, we introduce a random noise with higher intensity to accentuate the changes in low-frequency area, where visual details are typically absent. Then, in the denoising process, we utilize

the pre-trained SDXL model [38] as the denoiser, which is one of the most powerful open-source image diffusion models. The objective of denoising stage is not merely to eliminate noise but also to add high-quality details. To achieve this, we develop three gradient-based regularizers: image actuation, noise distribution, and adversarial degradation. These regularizers are designed to enhance the noise removal process by revising predicted noise, leading to the improvement of the overall image quality. Figure 1 illustrates the examples of the input images from HPDv2 dataset and the enhanced images by FreeEnhance.

In summary, we have made the following contributions: 1) The proposed FreeEnhance is shown capable of tuning-free strategy to improve the quality of the generated images; 2) The designs of content-consistent noising and three denoising corrections are unique; 3) FreeEnhance has been properly analyzed and verified through extensive experiments over HPDv2 dataset to validate its efficacy. With the good, due to the content-consistent capability, FreeEnhance can be readily applicable to enhance real images.

## 2 RELATED WORKS

### 2.1 Diffusion Models

Diffusion models [14, 20, 43, 48] have garnered attention for their remarkable generative quality and diversity in learning complex data distributions. They have been applied in various downstream tasks, including multimedia generations like text-to-image [42, 57], text-to-video [22], text-to-3D [39], and text-to-audio [17]. Diffusion models are devised to synthesize multimedia contents from an initial random noise by iterative denoising operations. Existing pixel-based diffusion models exhibit slow inference speeds and required substantial computational resources. Many creative researches devote into overcome this issue from applying discrete diffusion [5], using image tokens from VQ-VAE [18]. Among them, the Latent Diffusion Models (LDM) [41] operates the noising and denoising in a compressed latent space, effectively get out of this dilemma by striking a better trade-off between cost and generation quality. Subsequent improvements including attention mechanism [8], enhancing the architectures [13, 37] and prompt-tuning [15] have vigorously driven the development of diffusion models in ai-generated content. Moreover, as a basic paradigm, image-to-image

**Figure 3: An overview of our Tuning-Free Image Enhancement (FreeEnhance) framework. The process of FreeEnhance begins with an input image $x$, which undergoes a two-stream noising scheme to adaptively add noise into $x$. The creative steam adds strong noise which is then partially removed by a diffusion model with gradient-guided sampling (GGS), resulting $x_{t_0}^c$. And in the stable stream, light noise is attached with the input image using DDIM inversion strategy, obtaining $x_{t_0}^s$. Then $x_{t_0}^c$ and $x_{t_0}^s$ are adaptively blended according to the high/low frequency map $M_h/M_l$ produced by frequency filtering of $x$, resulting the noisy image $x_{t_0}$. Then, $x_{t_0}$ is fed into diffusion models which is constrained by three regularizers, which are devised from the perspectives of image acutance, noise distribution, and adversarial degeneration, in the denoising stage to produce the enhanced version of the input image.**

translation tasks also demonstrate the potential of using the diffusion model in style-transfer [52, 58], inpainting [28, 55] and image editing [24, 45].

## 2.2 Guidance in Diffusion Models

Guidance is a technique widely employed in the sampling process. It can be regarded as an extra update to the direction of sampling at each iteration and can modify the outputs after training by *guiding* with additional conditions, such as label [14], text [41]. Classifier guidance (CG) [14] improves quality and generates conditional samples by adding the gradient of a pre-trained class classifier. Similarly, CLIP guidance [35] utilizes similarity scores from a fine-tuned CLIP model [40]. To avoid training the classifier, classifier-free guidance (CFG) [21] drops the explicit classifier and models an implicit one by omitting the conditions with a certain probability during training. And others [6, 27, 29, 53] show that the gradient of also can be regard as the guidance. For instance, Composable Diffusion [27] adopts composed guidance from multi approximate energy. Contrastive Guidance [53] utilizes positive and negative prompt to build a contrastive pair and regards gradient of difference as guidance to guide sampling.

## 2.3 Image Enhancement for Human Preference

While sharing similarities with tasks like tradition image enhancement, image enhance on detail has been studied mainly on how to strike a better trade-off between detail and content consistency. Meanwhile, diffusion models have been implicitly endowed with image the enriching detail capability. Based on how this capability is built, we can broadly categorize existing studies into three classes. The first solution is the refinement model. SDXL refiner [38] train a separate LDM model in the same latent space. It can improve quality of detailed backgrounds. The second is the upscale-then-tile method. Recent studies [7, 16, 19] show that its capability to create details on local region and can keep content. Starting from upscaling an image, MultiDefusion [7] tile the image into a set of patches, then proposes fusing multiple diffusion paths on these

patches, resulting in high-resolution images. However, it suffers from the object repetition issues due to the prompt independently guiding the denoising of each patch. The third [3, 23] involves performing a secondary prediction on regions that are hard to generate during the denoising process. Self-attention guidance (SAG) [23] utilizes adversarial blurring on the regions of denoising model focused, then leverages the secondary predicted noise of blurred one to guide the sampling direction of the original one. It can effectively improve generation quality. Perturbed Attention Guidance (PAG) [3] introduce a perturbed attention layer which replaces the attention matrix with an identity matrix to improve quality.

## 3 METHOD

This section first reviews the diffusion models and the standard schemes of noising-and-denoising process for image enhancement without the consideration of content consistency (Section 3.1). Next, we describe how FreeEnhance properly add noise on the input image enabling creative generation while persevering attributes of contents (Section 3.2) in the noising stage. And then we introduce the noise removal using a diffusion model incorporated with three gradient-based terms. These terms, formulated from the perspective of acutance and visual quality of images, respectively, regularize the predicted noise and enhance image details in the denoising stage (Section 3.3). Figure 3 depicts the framework of our FreeEnhance.

## 3.1 Preliminary

Diffusion models create images by progressively removing noise through a series of denoising steps. This denoising process essentially reverses another process (i.e., noising process) that adds noise to an images in a pre-determined time-dependent manner. Specifically, given a timestep $t \in \{T, T-1, ..., 1\}$ and the noise $\epsilon_t$, the noisy image is created as $x_t = \alpha_t x + \sigma_t \epsilon_t$, where $x$ is the original image, $\alpha_t$ and $\sigma_t$ are parameters determined by the noise schedule and the timestep $t$. To perform the denoising process for image synthesis, a common choice for diffusion models is learning a neural network

$\epsilon_\theta$ that attempts to estimated the noise $\epsilon_t$, where $\theta$ is obtained by:

$$\arg\min_\theta \mathbb{E}_{t\sim\mathcal{U}(1,T),\epsilon_t\sim\mathcal{N}(0,\mathbf{I})} ||\epsilon_t - \epsilon_\theta(x_t;t,y)||^2, \quad (1)$$

and $y$ is an optional conditioning signal like text prompt. Once the model $\epsilon_\theta$ is trained, images can be generated by starting from noise $x_T \sim \mathcal{N}(0,\mathbf{I})$ and then alternating between noise estimation and noisy image updating:

$$\hat{\epsilon}_t = \epsilon_\theta(x_t;t,y), \ x_{t-1} = update(x_t,\hat{\epsilon}_t,t), \quad (2)$$

where the updating can be performed by DDPM [20], DDIM [47], DPM [46] or other sampling algorithms. Using the reparameterization trick [20], we can further obtain an intermediate reconstruction of $x_0$ at a timestep $t$, denoted as $\hat{x}_{t\to 0}$.

To improve the realism and faithfulness to the condition in generated images, SDEdit [32] and SDXL [38] utilize a noising-and-denoising process, which first add random noise corresponding to the timestep $t_0$ into the input image and then subsequently denoises the resulting image. The hyper-parameter $t_0$ can be tuned to tradeoff between consistency and creativity: with a smaller $t_0$ leading to a more content-consistent but less local detailed generated image. This approach treats every region of the input image the same. It adds random noise with the same intensity at timestep $t_0$ across the entire image, disregarding the varying needs of different areas. Some regions might benefit from creatively introduced details, while others might require meticulous preservation of existing content. As the result, the naive noising-and-denoising process struggles to find a balance, either over-editing images or leaving them lacking in detail.

## 3.2 Noising Stage

To alleviate these issues, our FreeEnhance tailors the noising process following an intuitive idea: High-frequency areas, rich in edges and corners, should receive lighter noise to safeguard their original patterns. Conversely, low-frequency regions are expected to be exposed to stronger noise, promoting creative detail generation and refinement. Considering the assumption of diffusion models that all regions/pixels of noisy images share the same noise distribution (i.e., the intensity of noise), we propose a two-stream noising scheme to adaptively add noise into the original image. The creative stream involves higher intensity of noise to enrich image details and the stable stream introduces weaker noise to maintain content fidelity.

For the creative stream which is divised for creative detail generation, a random noise corresponding to timestep $T$ is added into the input image $x$, obtaining the noisy image $x_T^c = \alpha_T x + \sigma_T \epsilon_T$. Then a diffusion model is utilized to iteratively denoising $x_T^c$ till the timestep $t_0$ and obtain $x_{t_0}^c$. Although the variant of the input image is encouraged during the denoising, we still need to align the structural elements (often determined by edges and corners located in high-frequency regions) of $x_{t_0}^c$ with those of the input image $x$. Thus we utilize the gradient-guided sampling [9, 10, 12] to introduce conditioning on auxiliary informamtion [9] for the denoising process $x_T^c \to x_{t_0}^c$ in this noising stream. The gradient-guided sampling utilizes guidance generated from pre-defined energy functions $g(x_t;t,y)$ to altering the update direction $\hat{\epsilon}_t$:

$$\hat{\epsilon}_t = \epsilon_\theta(x_t;t,y) + \lambda\sigma_t \nabla_{x_t} g(x_t;t,y), \quad (3)$$

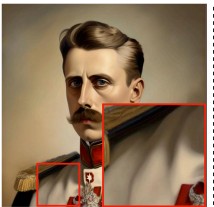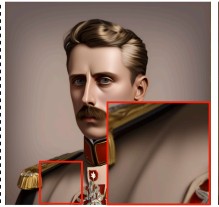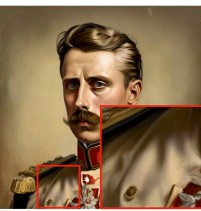

| Input | Output w/o calibration | Output with calibration |

**Figure 4: Comparison between images generated from composited noisy image with and without the distribution calibration. The color shift/fading can be observed on the image generated without the calibration.**

or revise the sampling result $x_{t-1}$:

$$x_{t-1}^* = x_{t-1} - \lambda\nabla_{x_t} g(x_t;t,y), \quad (4)$$

where $\lambda$ is the weight of the additional guidance. Here we define the energy function $g(x_t;t,y) = M_h||x - \hat{x}_{t\to 0}||^2$ for gradient-guided sampling, where $M_h$ is a binary map obtained by high-pass filtering [51] on the input image to identify high-frequency regions.

For the stable stream, we employ the DDIM inversion strategy [33] to add noise into $x$ and obtain the noisy image $x_{t_0}^s$. Such an approach ensures that the contents in $x$ are able to be reconstructed from $x_{t_0}^s$ with high fidelity when we utilized a deterministic sampling algorithm like DDIM.

Once two noisy images $x_{t_0}^c$ and $x_{t_0}^s$ are produced by the creative and stable noising streams, we adaptively blend the two noisy image according to the frequency of image regions. For the high-frequency image regions localized by the map $M_h$, we directly involve $x_{t_0}^s$ to maintain the the content structure, resulting $x_{t_0}^h = M_h x_{t_0}^s$. And for the low-frequency regions which are marked by $M_l = 1 - M_h$, we conduct an alpha-compositing for $x_{t_0}^c$ and $x_{t_0}^s$ using a tradeoff parameter $\tau$:

$$x_{t_0}^l = M_l(\tau x_{t_0}^s + (1-\tau)x_{t_0}^c). \quad (5)$$

However, the distribution of weighted average of two noisy images is $\mathcal{N}(\alpha_t x, \frac{\sigma_t^2}{2\tau^2 - 2\tau + 1}\mathbf{I})$, which violates the hypothesized prior distribution $\mathcal{N}(\alpha_t x, \sigma_t^2 \mathbf{I})$ of the diffusion process, resulting suboptimal image generation (e.g., over-smooth surface). To mitigate this, we rescale the composited noisy image using a scale factor $1/\sqrt{2\tau^2 - 2\tau + 1}$ to calibrate the distribution. Figure 4 demonstrates the comparison between images generated from composited noisy image with and without the distribution calibration.

## 3.3 Denoising Stage

With the noisy image produced by the noising stage of FreeEnhance, we subsequently conduct the denoising process and present three target properties as constraints to regularize the predict noise and/or revise the updated noisy images from the aspects of image acutance and noise distribution. Such constraints are formulated from a score-based perspective [4] of diffusion models and leverage the capability of diffusion models which can adapt outputs by guiding the sampling process.

**Acutance Regularization.** In photography, acutance refers to the perceived sharpness associated with the edge contrast of an image [30]. Owing to characteristics of the human visual system,

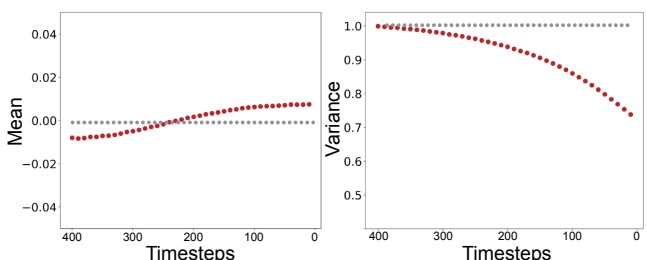

**Figure 5: The statistics of the noise $\epsilon_\theta(x_t; t, y)$ predicted by a diffusion model. Given the noisy image from the noising stage of FreeEnhance, the red scatters is estimated during the denoising process using SDXL and the gray ones represent the ideal values across different timesteps.**

images with higher acutance tend to appear sharper, despite the fact that an increase in acutance does not have to enhance actual resolution of images. Here we utilize the acutance of $\hat{x}_{t\to0}$, which is the intermediate reconstruction of $x_0$ at the timestep $t$, to regularize the denoising. Specifically, we utilize the Sobel kernel to estimate the magnitude of the derivative of brightness concerning spatial variations of $\hat{x}_{t\to0}$, denoted as $\mathcal{F}_{acu}(\hat{x}_{t\to0})$. To encourage a higher acutance, the objective of acutance regularization is:

$$\mathcal{L}_{acu} = -\frac{1}{HW} \sum_{i=0, j=0}^{H,W} \mathcal{F}_{acu}(\hat{x}_{t\to0})_{(i,j)} , \quad (6)$$

where $H, W$ represent the spatial size of the noisy image and $(i, j)$ are the indices of the spatial element. This formulation assumes that all spatial locations in the generated images are intended to be 'sharp'. But in practice, emphasizing all the edges/corners of the input image may introduce unpleasant structures in the flat regions (e.g., sky and metal surfaces) and intricate regions (e.g., trees and bushes), impacting human preferences. To tackle this issue, we extend the formulation in Eq. 6 with a binary indicator $V(\cdot)$:

$$\mathcal{L}_{acu} = -\frac{1}{HW} \sum_{i=0, j=0}^{H,W} V(\mathcal{F}_{acu}(\hat{x}_{t\to0})_{(i,j)})\mathcal{F}_{acu}(\hat{x}_{t\to0})_{(i,j)} . \quad (7)$$

where $V(\cdot) = 1$ when the input value falls within the 35th and 65th percentiles of $\mathcal{F}_{acu}(\hat{x}_{t\to0})$. Accordingly, our acutance regularization introduces additional details into the images while minimizing unpleasant structures, enhancing the overall generation quality.

**Distribution Regularization.** Considering the inevitability of generalization error, the noise predicted by diffusion models $\epsilon_\theta(x_t; t, y)$ may not fully follow a gaussian distribution $\mathcal{N}(0, \mathbf{I})$, particularly when we directly utilize a diffusion model to generate images from the composited noisy image produced in our noising stage. To validate this assumption, we analyze more than 3,000 images and summarize the distribution of predicted noise during the denoising process in Figure 5. We observe that the mean values of the predicted noise approach zero across different timesteps during denoising, while the difference between the actual variance values and 1 is nonnegligible when the timestep is large. Building upon this intuition, we regularize the denoising process via punishing the gap of distribution:

$$\mathcal{L}_{dist} = ||1 - \mathcal{F}_{var}(\epsilon_\theta(x_t; t, y))||_2 , \quad (8)$$

where $\mathcal{F}_{var}$ caluates the variance of the predicted noise.

**Adversarial Regularization.** Motivated by the self-attention guidance for diffusion models [23], we incorporate an adversarial regularization for the denoising stage of FreeEnhance to avoid generating blurred images. Specifically, we define the $\mathcal{F}_{blur}$ as a gaussian blur function and devise the objective as follws:

$$\mathcal{L}_{adv} = ||\hat{x}_{t\to0} - \mathcal{F}_{blur}(\hat{x}_{t\to0})||_2 . \quad (9)$$

**Regularizating the Denoising.** With the help of the three regularizations, we additionally insert a revising step at the end of updating in each denoising iteration. Specifically, the sampling result $x_{t-1}$ in each denoising operation is altered by $x_{t-1}^*$:

$$x_{t-1}^* = x_{t-1} - \rho_{acu} \nabla_{x_t} \mathcal{L}_{acu} - \rho_{dist} \nabla_{x_t} \mathcal{L}_{acu} - \rho_{adv} \nabla_{x_t} \mathcal{L}_{adv} , \quad (10)$$

where $\rho_{acu} = 4$, $\rho_{dist} = 20$, and $\rho_{adv} = 0.3$ are the tradeoff parameters determined through experimental studies.

## 4 EXPERIMENTS

We empirically verify the merit of FreeEnhance for tuning-free image enhancement on the public dataset HPDv2 [54] following the evaluation protocol [11, 26] in terms of the quantitative metrics and qualitative human preference. We first introduce the dataset, quantitative metrics, baseline approaches, and implementation details of our FreeEnhance (Section 4.1). Next, we elaborate the comparisons between FreeEnhance and baselines on both quantitative and qualitative results (Section 4.2), followed by the comparison to Magnific AI (Section 4.3). Finally, we analyze the designs in our FreeEnhance via ablation studies (Section 4.4).

### 4.1 Experimental Settings

**Dataset.** Human Preference Dataset v2 (HPDv2) [54] is a large-scale dataset of human preferences for images generated from text prompts. It comprises 798,090 human preference choices on 433,760 pairs of images. HPDv2 provides a set of evaluation prompts that involves testing a model on a total of 3,200 prompts, evenly divided into 4 styles: Animation, Concept-Art, Painting, and Photo. For each type of evaluation prompt, HPDv2 provides the corresponding benchmark images generated by various mainstream text-to-image generative models. Here we exploit the group of benchmark images generated by SDXL-Base-0.9 as the inputs of image enhancement approaches to validate the merit of our proposal.

**Metrics.** Non-reference image quality assessment (NR-IQA), also called blind IQA, is a metric for evaluating the quality of an image without needing its pristine version for comparison. We employ three kinds of NR-IQA metrics for quantitative evaluation, including MANIQA [56], CLIPIQA+ [50] and MUSIQ [25]. Since each of the three metrics has multiple publicly available versions, involving fine-tuning from different datasets (e.g., KADID and KonIQ) or employing different models (e.g., ResNet and ViT), here we evaluate image quality using multiple metrics from MANIQA (3 versions), CLIPIQA (3 versions), and MUSIQ (2 versions). Human Preference Score v2 (HPSv2) [54] is a scoring model that trained on the HPDv2 dataset to predict human preferences on the generated images. We utilize the HPSv2 to score the images before/after enhancement to verify the quality improvement.

**Implementation Details.** We use the base model of Stable Diffusion XL (SDXL-base) implemented in HuggingFace Transformer

**Table 1: Quantitative comparisons on HPDv2 benchmark images generated by SDXL-Base-0.9 for image enhance. We mark the best results in bold. † means run with prompts. For MANIQA and MUSIQ, we report their sub-versions fine-tuned on different datasets. For CLIPIQA+, we report its sub-versions with different backbone. The * denotes the sub-version fine-tuned with both positive and negative prompts. The HPSv2 score adopted v2.1 version.**

| Method | MANIQA ↑ | | | CLIPIQA+↑ | | | MUSIQ↑ | | HPSv2↑ |
|---|---|---|---|---|---|---|---|---|---|
| | KonIQ | KADID | PIPAL | ResNet50 | ResNet50* | ViT-L | KonIQ | SPAQ | |
| SDXL-base [38] | 0.3609 | 0.5821 | 0.5721 | 0.5688 | 0.4074 | 0.4267 | 63.7948 | 62.1716 | 27.39 |
| SDXL-refiner [38] | 0.3305 | 0.6006 | 0.5674 | 0.5671 | 0.3658 | 0.3636 | 61.7321 | 60.8410 | 27.27 |
| SAG [23] | 0.3933 | 0.6088 | 0.6154 | 0.6311 | 0.4450 | 0.4583 | 66.7950 | 65.1907 | 28.48 |
| Fooocus [2] | 0.4180 | 0.6359 | 0.6096 | 0.6130 | 0.4612 | 0.4667 | 68.0095 | 65.7813 | 28.31 |
| DemoFusion† [16] | 0.2747 | 0.5414 | 0.5129 | 0.5085 | 0.3424 | 0.3607 | 56.5470 | 58.5090 | 27.87 |
| FreeU [45] | **0.4194** | 0.6272 | 0.5938 | 0.6189 | 0.4649 | 0.4703 | 68.0083 | 66.2340 | 28.88 |
| FreeEnhance | 0.4122 | **0.6611** | **0.6332** | **0.6535** | **0.4929** | **0.4901** | **68.3928** | **66.8653** | **29.32** |

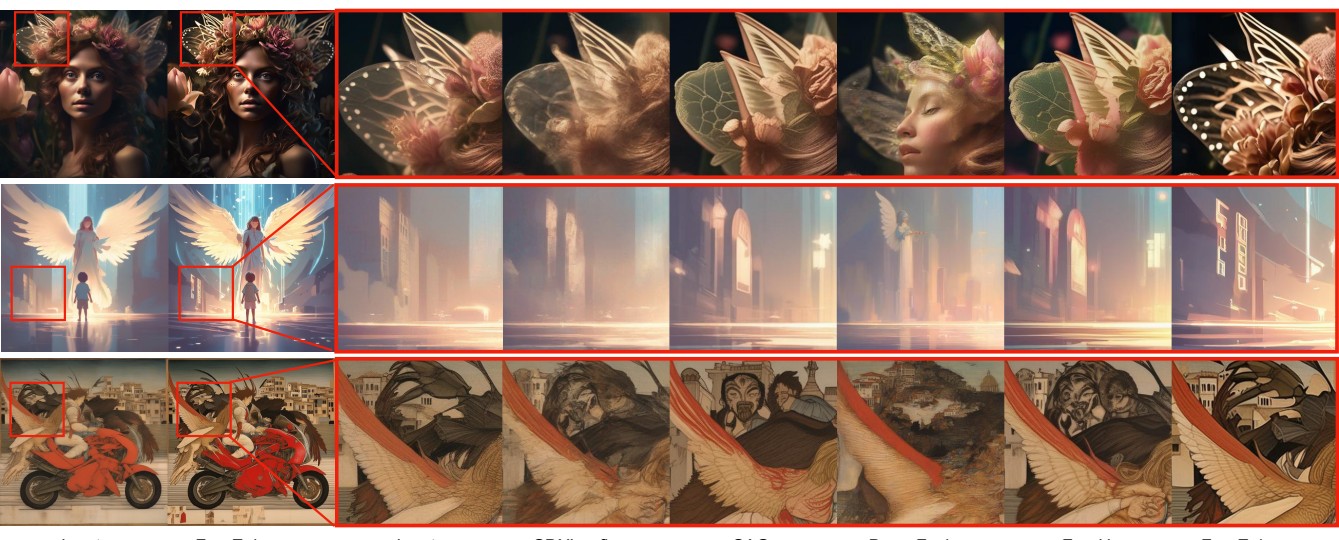

| Input | FreeEnhance | Input | SDXL refiner | SAG | DemoFusion | FreeU | FreeEnhance |

**Figure 6: Quantitative comparisons of images enhanced by different approaches on HPDv2 benchmark. The regions in red boxes are presented in zoom-in view to ease the comparison.**

and Diffuser libraries [49] as the diffusion model for image enhancement, unless otherwise stated. Hence, the noising-and-denoising process is conducted in the latent feature space. The high/low frequency regions of the input images are recognized by the high/low filtering proposed in DR2 [51]. The resolution of images before and after enhancement is $1,024 \times 1,024$ and the original prompts of the benchmark images from HPDv2 are not involved by our FreeEnhance. During the noising-and-denoising process of FreeEnhance, the inference steps is set as 100, with a guidance scale of 1.0. The hyper-parameter $t_0$ that indicates the strength of attached noise is set as 500. All experiments are conducted on NVIDIA RTX 3090 GPUs and Intel Xeon Gold 6226R CPU.

## 4.2 Performance Comparison

**Quantitative Results.** We compare our FreeEnhance with several open-source off-the-shelf approaches in terms of three groups of NR-IQA metrics and one human preference metric in Table 1. All the mentioned baselines are grouped into three directions: plain noising-and-denoising with diffusion model (SDXL-base and

SDXL-refiner [38]), upscale-then-tile operation (DemoFusion [16]) and sampling with guidance scheme (SAG [23], Fooocus [2], and FreeU [45]). Note that all methods, except for "SDXL-refiner", use the pre-trained diffusion model SDXL-base. "SDXL-refiner" utilizes custom weights. In general, our FreeEnhance approach consistently achieves better image quality compared to these baselines. Notably, FreeEnhance attains a score of 29.32 on the HPSv2 metric without any diffusion model parameter tuning. Compared to the baseline of SDXL-base, the SDXL-refiner produces unsatisfactory image enhancement results due to the relatively high intensity of the attached noise which is constrained on the first 200 (discrete) noise scales during the training of diffusion model. Benefitting from the self-attention guidance employed during noise removal, SAG and Fooocus exhibit better generation quality and have performance gain on NR-IQA metrics and the HPSv2 scores (28.48/28.31 vs. 27.39). DemoFusion has decent performances on both NR-IQA metrics and HPSv2. We speculate that this may be the result of the employed shifted crop sampling with delated sampling which introduces unnatural local textures. FreeU conducts the denoising in a frequency decoupled manner, which leads to better enhancement results on

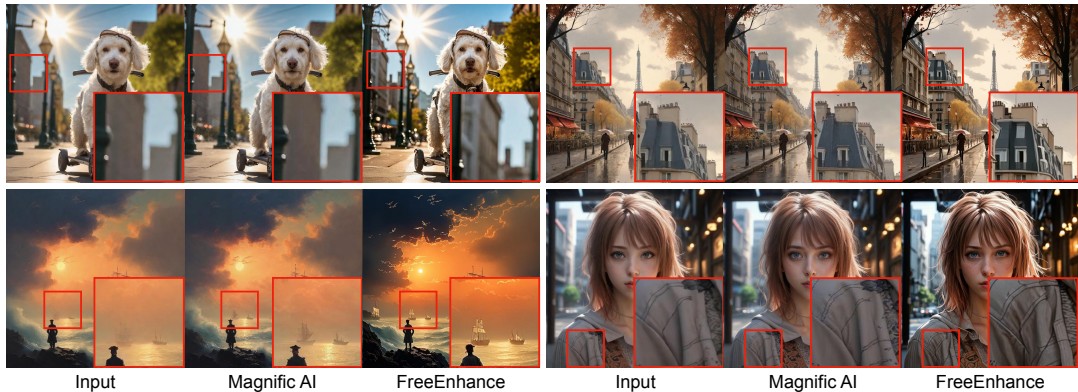

**Figure 7: Quantitative comparisons of images enhanced by Magnific AI and our FreeEnhance. The regions in red boxes are presented in zoom-in view to ease the comparison.**

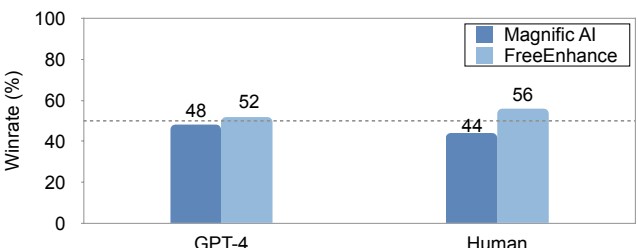

**Figure 8: Comparisons between FreeEnhance and Magnific AI with regard to GPT-4 and human preference ratios.**

**Table 2: Ablation study on each design in FreeEnhance on HPDv2 benchmark.**

| Noising Stage | Denoising Stage | | | HPSv2 |
| --- | --- | --- | --- | --- |
| | Distribution | Acutance | Adversarial | |
| ✖ | ✖ | ✖ | ✖ | 27.39 |
| ✔ | ✖ | ✖ | ✖ | 28.63 |
| ✔ | ✔ | ✖ | ✖ | 28.71 |
| ✔ | ✔ | ✔ | ✖ | 28.92 |
| ✔ | ✔ | ✔ | ✔ | **29.32** |

both NR-IQA and HPSv2. FreeEnhance, which simultaneously considers both ways to add and remove noise to the input images for quality improvement, obtains the highest HPSv2 score 29.32, surpassing the best competitor FreeU by 0.44 in HPSv2. The results demonstrate the effectiveness of frequency-adaptive noise addition and regularized denoising for image enhancement by diffusion models.

**Qualitative Results.** We then visually examine the enhancement quality of our proposal by comparing FreeEnhance with four baselines: SDXL-refiner, SAG, DemoFusion, and FreeU on three input images. Figure 6 shows the qualitative results of the enhanced images. To better illustrate the image details, we provide zoom-in views of image patches. Overall, all the approaches successfully modify the input images, and our FreeEnhance creates the most plausible local textures and details in the images while maintaining good content consistency between the input images and the enhanced ones. Taking the image in the first row as an example, FreeEnhance nicely provides more detailed structures, clear boundaries, and realistic material for the headwear, while preserving its shape and characteristics. In contrast, the SDXL-refiner fails to reconstruct the input image, resulting in a corrupted outcome. SAG and FreeU produce moderate modifications and add several detail structures, but still lose the sparkling points at the left side of the headwear. DemoFusion dramatically changes the headwear to a human face, which is not desired in image enhancement.

### 4.3 Comparison with Magnific AI

Figure 7 presents a comparative analysis of image enhancement results between our FreeEnhance and Magnific AI, renowned for its

advanced image enhancement capabilities. In the first case, Magnific AI falls short in providing additional detailed structure for the building situated on the left side of the image. Conversely, our FreeEnhance seamlessly enhances the visual quality and realism of the external facades of the building, while adeptly preserving both the content and the depth of field.

We further conduct a human study to examine whether the enhanced images from FreeEnhance better conform to human preferences than that given by Magnific AI. Specifically, we randomly sample 100 prompts from HPDv2 and generate $1,024 \times 1,024$ images using SDXL-base. We recruited 50 evaluators, including 25 males and 25 females, with diverse educational backgrounds and ages. Each evaluator was tasked to select the preferred image from two options generated by different paradigms but originating from the same original image. Evaluators were encouraged to choose the image that best satisfied their preferences. We also conduct the same evaluation using the GPT-4. Figure 8 illustrates the preference ratios. Overall, FreeEnhance achieves competitive results both on human and GPT-4 study.

### 4.4 Experimental Analysis

**Ablation Study.** We investigate how each design in our FreeEnhance influences the visual quality of the enhanced images. Table 2 details the performances (i.e., HPSv2 scores) across different ablated runs of FreeEnhance. We start from a basic noising-and-denoising scheme using the SDXL-base diffusion model, which achieves 27.39 of the HPSv2 score. Next, by solely using the noising stage of FreeEnhance which adaptively add light noise on high-frequency regions and strong noise on low-frequency regions, we observe a clear

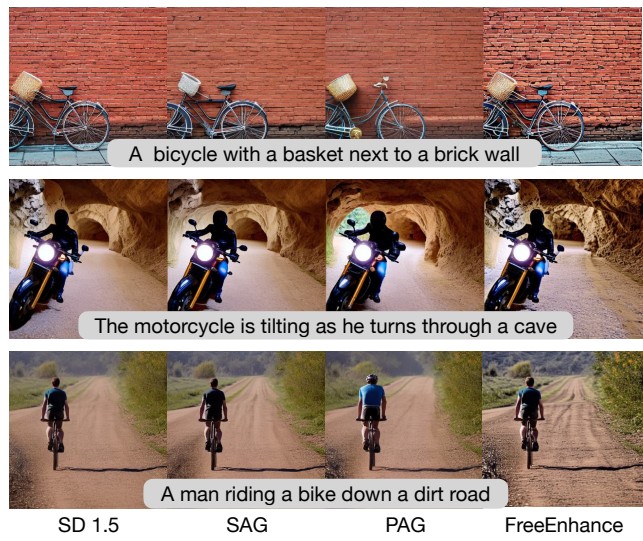

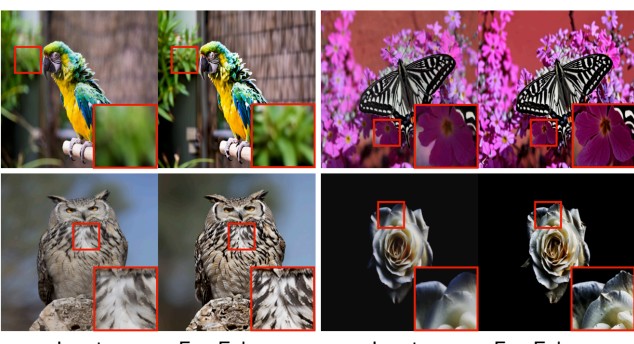

Figure 10: Examples of natural images which are enhanced by FreeEnhance.

A bicycle with a basket next to a brick wall

The motorcycle is tilting as he turns through a cave

A man riding a bike down a dirt road

| SD 1.5 | SAG | PAG | FreeEnhance |

Figure 9: Qualitative comparisons of images synthesized using various denoising approaches in the text-to-image scenario, using prompts from HPDv2 benchmark.

Table 3: Comparisons of HPSv2 scores of images produced by different diffusion models with/without FreeEnhance.

| Model | w/o FreeEnhance | w/i FreeEnhance |
|---|---|---|
| SDXL-base | 27.39 | **29.32** |
| SDXL-refiner | 27.27 | **29.15** |
| DreamshaperXL | 29.52 | **30.06** |

Table 4: Comparison of HPSv2 scores for images synthesized using various denoising approaches in the text-to-image scenario, employing SD 1.5 on the HPDv2 benchmark.

| Method | SD 1.5 | SAG [23] | PAG [3] | FreeEnhance |
|---|---|---|---|---|
| HPSv2 ↑ | 24.61 | 24.76 | 25.02 | **25.26** |

performance boost. We then leverage the three regularizers in the denoising stage in turn. The HPSv2 score is consistently boosted up by three regularizers and finally reaches 29.32.

**Effect of the diffusion models.** To investigate the impact of the diffusion model on image enhancement, we utilize three diffusion models: SDXL-base, SDXL-refiner, and DreamshaperXL [1], to execute the noising-and-denoising process with and without our proposed FreeEnhance. Table 3 summarizes the HPSv2 scores of the images produced by various diffusion models with/without FreeEnhance. The results constantly verify that FreeEnhance generates superior images regardless of the model used.

## 4.5 Applications

To assess the generalization capability of FreeEnhance, we conduct additional experiments under different two scenarios.

**Text-to-Image Generation.** In view that the denoising stage in FreeEnhance can be simply applied to the Gaussian random noise for image generation without the reference image, we can perform text-to-image generation using our FreeEnhance. To validate the capability of FreeEnhance, we synthesize images for the prompts in HPDv2 benchmark using the stable diffusion 1.5 with different denoising approaches. The scale of the classifier-free guidance is fixed as 7.5. Table 4 details the comparison results. FreeEnhance achieves the highest HPSv2 score (25.26), surpassing both the vanilla denoising schemes of SD 1.5 (24.61) and the advanced approaches

SAG (24.76) and PAG (25.02). We further showcase three examples in Figure 9. Overall, all four methods correctly align the prompt, and FreeEnhance presents superior visual quality, with the evidence of the clearer depiction of bricks on the wall (1st row), and the more realistic representation of dirt and gravel blocks on the road (2nd and 3rd rows). These results again highlight the generalization capability of FreeEnhance's designs.

**Natural Image Enhancement.** Here we empirically evaluate the capability of FreeEnhance on natural images. We select images from the LAION-5B dataset [44] and enhance their quality using our FreeEnhance. Figure 10 showcases four pairs of enhancement results. For instance, the leaves of the tree in the first case become clearer after enhancement. The results indicate that FreeEnhance is well-suited for refining natural images.

## 5 CONCLUSION

We have presented FreeEnhance for image enhancement by exploiting the off-the-shelf text-to-image diffusion models. Particularly, FreeEnhance formulates image enhancement as a two-stage process, which firstly attaches random noise to the input image, followed by noise reduction through the diffusion model. In the noising stage, we devide the input image into high/low frequency regions, adding light/strong random noise to preserve existing content structures while enhancing visual details. In the denoising stage, we introduce three gradient-based regularizations to revise the predicted noise, leading to the improvement of the overall image quality. The results on the image generation benchmark demonstrate superior visual quality and human preference over state-of-the-art image enhancement approaches. Furthermore, the FreeEnhance model is readily applicable to enhance natural images taken by the end users, enabling a wide range of real-life applications.

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
