# OpenReview forum: "FreeEnhance: Tuning-Free Image Enhancement via Content-Consistent Noising-and-Denoising Process"
_acmmm.org/ACMMM/2024/Conference — MM2024 Poster_

### Official Review · Reviewer_eusc · 2024-05-23

**Rating:** 4
**Confidence:** 2

**Summary:**

This paper proposes a novel framework, termed FreeEnhance, for content-consistent image enhancement.  A two-stage process is leveraged: noising and denoising. Noising stage aims at adding lighter noise to high-frequency areas to keep patterns like edges and corners, while denoising stage aims to use three target properties to control the predicted noise, improving image sharpness and quality.

**Strengths:**

1.	There are adequate comparative experiments to validate the effectiveness of the proposed method.
2.	The proposed method has excellent visual superiority over other methods.

**Limitations:**

1.	Paper writing contains a lot of long sentences that make comprehension more difficult, specifically in Abstract.
2.	The key process of “noising-and-denoising” is not novel enough.
3.	The inference speed of the proposed method is unclear.

**Suitability:**

3

---

### Official Review · Reviewer_u3bz · 2024-05-24

**Rating:** 4
**Confidence:** 2

**Summary:**

In this paper, a non-tuning framework called FreeEnhance is proposed to improve the quality of generated images through a content-consistent noise addition and de-noising process. The framework is divided into two stages: noise addition stage and noise removal stage. In the noise addition phase, the authors propose a two-stream noise addition scheme in which lighter noise is added to the high frequency region to preserve edge and corner details, while heavier noise is added to the low frequency region to enrich the details of the smooth region. In the denoising stage, the author uses the pre-trained diffusion model for denoising, and proposes three gradient-based regularization terms: image sharpness, noise distribution and counter degradation to further improve the visual quality of the generated image.

**Strengths:**

1. Innovative points: A consistent two-stream noise addition scheme is proposed, which can retain the details of the high-frequency region while enriching the details of the low-frequency region, solving the tradeoff problem between creativity and content consistency in the previous noise addition process.
2. Effectiveness: A large number of experiments have shown that FreeEnhance is superior to existing image enhancement models and even better than commercial solutions in terms of quantitative indicators and human preference assessment.
3. Wide applicability: Due to the advantages of content consistency,FreeEnhance can not only enhance the quality of the generated image, but also be applied to enhance the real image.

**Limitations:**

1. Lack of more in-depth analysis of the noise addition scheme: The motivation and method of the double-flow noise addition scheme in this paper are not thoroughly described, and more in-depth theoretical analysis and empirical research are lacking.
2. Complexity of regularization terms: The three regularization terms proposed are relatively complex and require certain parameters and computing resources. The computational cost of these regularization terms is not analyzed in this paper.
3. Lack of noise level analysis: The paper does not carry out in-depth comparative analysis of different levels of noise addition schemes.

**Suitability:**

3

---

### Official Review · Reviewer_BFWX · 2024-05-26

**Rating:** 5
**Confidence:** 3

**Summary:**

To enrich plentiful details while preserving the visual appearance of key content, this paper proposes a new framework (FreeEnhance) for content-consistent image enhancement. Specifically, FreeEnhance first adds random noise to the input image and then capitalizes on Latent Diffusion Models to denoise and enhance the image details. Then, FreeEnhance is devised to add lighter noise to the region with higher frequency to preserve the high-frequent patterns. The authors claim that the experimental results demonstrate the state-of-the-art performance and effectiveness of their method.

**Strengths:**

1. This paper proposes a tuning-free method for content-consistent image enhancement.
2. The proposed FreeEnhance can be generally applied to real images for its content-consistent capability.
3. This paper is well-written and presented with clear motivation. The technology of this paper also generally sounds good.

**Limitations:**

1. I wonder about the impact of the intensity of noise on the performance of the proposed FreeEnhance during the training stage.
2. The inference time for every image is unclear by using the proposed FreeEnhance.

**Suitability:**

3

---

### Meta-Review · Area_Chair_QaW6 · 2024-07-02

**Recommendation:** Accept (Poster)
**Confidence:** 5

**Metareview:**

All reviewers hold positive attitudes towards this work. AC agrees with each reviewer and decides to accept this work. The authors are suggested to consider all suggestions in their final version.